# The Habitual Diet of Dutch Adult Patients with Eosinophilic Esophagitis Has Pro-Inflammatory Properties and Low Diet Quality Scores

**DOI:** 10.3390/nu13010214

**Published:** 2021-01-13

**Authors:** Marlou L. A. de Kroon, Simone R. B. M. Eussen, Bridget A. Holmes, Lucien F. Harthoorn, Marijn J. Warners, Albert J. Bredenoord, Bram D. van Rhijn, Mylene van Doorn, Berber J. Vlieg-Boerstra

**Affiliations:** 1Department of Health Sciences, University Medical Center Groningen, University of Groningen, 9713 GZ Groningen, The Netherlands; m.l.a.de.kroon@umcg.nl; 2Danone Nutricia Research, 3508 TC Utrecht, The Netherlands; simone.eussen@danone.com (S.R.B.M.E.); lucien.harthoorn@clasado.com (L.F.H.); mylenevandoorn@hotmail.com (M.v.D.); 3Global Nutrition Department, Danone Nutricia Research, 91120 Palaiseau, France; bridgetaholmes@gmail.com; 4Department of Gastroenterology & Hepatology, Sint Antonius Hospital, 3435 CM Nieuwegein, The Netherlands; m.warners@antoniusziekenhuis.nl; 5Department of Gastroenterology & Hepatology, Academic Medical Center, 1105 AZ Amsterdam, The Netherlands; a.j.bredenoord@amsterdamumc.nl; 6Department of Dermatology & Allergology, University Medical Center, 3584 CX Utrecht, The Netherlands; b.d.van.rhijn@gmail.com; 7Division of Human Nutrition, Wageningen University, 6708 WE Wageningen, The Netherlands; 8Department of Pediatrics, OLVG Hospital, 1091 AC Amsterdam, The Netherlands; 9Department of Nutrition & Dietetics, Hanze University of Applied Sciences, 9747 AS Groningen, The Netherlands

**Keywords:** diet, diet quality, food intakes, adults, eosinophilic esophagitis

## Abstract

We determined the nutritional adequacy and overall quality of the diets of adult patients with eosinophilic esophagitis (EoE). Dietary intakes stratified by sex and age were compared to Dietary Reference Values (DRV). Overall diet quality was assessed by two independent Diet-Quality-Indices scores, the PANDiet and DHD-index, and compared to age- and gender-matched subjects from the general population. Lastly, food and nutrient intakes of EoE patients were compared to intakes of the general population. Saturated fat intake was significantly higher and dietary fiber intake significantly lower than the DRV in both males and females. In males, the DRV were not reached for potassium, magnesium, selenium, and vitamins A and D. In females, the DRV were not reached for iron, sodium, potassium, selenium, and vitamins A, B2, C and D. EoE patients had a significantly lower PANDiet and DHD-index compared to the general population, although the relative intake (per 1000 kcal) of vegetables/fruits/olives was significantly higher (yet still up to 65% below the recommended daily amounts) and alcohol intake was significantly lower compared to the general Dutch population. In conclusion, the composition of the habitual diet of adult EoE patients has several pro-inflammatory and thus unfavorable immunomodulatory properties, just as the general Dutch population, and EoE patients had lower overall diet quality scores than the general population. Due to the observational character of this study, further research is needed to explore whether this contributes to the development and progression of EoE.

## 1. Introduction

Eosinophilic esophagitis (EoE) is a chronic, immune-mediated condition of the esophagus affecting both adults and children [1,2]. Although the exact pathogenic mechanism of the disease remains to be elucidated, exposure of the esophagus to foods and aeroallergens in genetically predisposed subjects is suggested to influence the development of the disease [3,4,5,6]. Foods most commonly implicated in EoE are cow’s milk, egg, soy, wheat or gluten, legumes, fish/shellfish and nuts/peanuts [4,7,8,9]. Disease remission can be accomplished by using topical corticosteroids or proton-pump inhibitors (PPIs), or by the dietary elimination of food allergens proven to trigger EoE [1,6,9,10]. Both elimination diets as well as elemental diets have shown remarkable clinical and histological responses in adults with EoE [9,11,12]. Additionally, many individuals with EoE adapt their eating behavior by chewing their foods very carefully, taking small bites, consuming more beverages with a meal, and eating foods with a softer consistency [13,14].

Besides the fact that food allergens can trigger an allergic reaction in EoE, in a previous study it was hypothesized that healthy and immune-enhancing nutrition may either be protective against the development of EoE or, in contrast, unhealthy nutrition may facilitate the development of food allergy through its impact on inflammation and esophageal mucosal integrity [15]. However, hardly any data have been published about dietary composition in EoE patients. In a cross-sectional study in adults with EoE, we recently showed that high intakes of dietary fiber, iron, fermented dairy (buttermilk, Lactobacillus rhamnosus GG-containing yoghurt drink), dairy, pasta/rice, and soy were negatively related to the degree of inflammation and positively related to mucosal integrity in EoE patients. In contrast, phosphorus (abundant in animal-based foods), high intakes of omega-6 rich oil (sunflower and/or stir fry oil), and total added fat had a positive relationship with inflammation and a negative relationship with mucosal integrity [15].

To further study the nutritional quality of the habitual diet in EoE, the present study aims were threefold: (1) to compare the nutrient intakes of Dutch adult patients with EoE with Dutch Dietary Reference Values (DRV) by age and sex; (2) to compare the overall diet quality of EoE patients with diet quality of the general Dutch population by using Diet Quality Indices (DQI), and (3) to compare the food and nutrient intakes of EoE patients with intakes of the general Dutch population. The study was performed prior to any elimination diets other than self-imposed dietary measures. DQI provide a single score for a combination of foods and/or nutrients, based on current nutrition knowledge established in food or nutrient based dietary guidelines [16,17,18]. Since foods and nutrients are consumed in combination and may interact with each other, DQI can be used when investigating diet–disease associations [16,17,18].

## 2. Materials and Methods

### 2.1. Study Population

Adult patients diagnosed with active EoE (≥15 eosinophils/high power field (HPF) and symptoms of esophageal dysfunction), who participated in two trials (Trialregister.nl NTR4052 and NTR4892) performed in the Academic Medical Center, Amsterdam, the Netherlands, were included between 2013 and 2015. Patients participating in Trial NTR4052 followed an allergen-microarray-guided dietary intervention as treatment of EoE [19]. Patients participating in Trial NTR4892 were treated with an elemental diet for four weeks [9,20]. From these two studies, baseline data, i.e., before starting any (elimination) diet other than self-imposed dietary measures, were used in the present study. Information on atopic status and food avoidance was collected by medical and allergy focused diet history.

### 2.2. Dietary Intake Assessment

All patients completed a 3-day non-weighed food diary (2 working days and 1 weekend day) as described previously (online supporting information in [15]). In short: Patients recorded amounts of foods and drinks in detail as well as dietary supplements (type, yes/no, infrequently). Portion size was assessed using household measures. The average of the 3-day diaries (without supplements) was calculated using the Dutch NEVO-online Food Composition Database [21] and Evry software [22] for energy (kcal) and 32 nutrients (Appendix A). Additionally, all foods consumed were allocated to one of 38 food groups, based on the Dutch National Food Consumption Survey (DNFCS) [23] (see Appendix A).

#### 2.2.1. Dutch Dietary Reference Values

The 3-day mean habitual intake of nutrients, stratified for age group and gender, was compared to the Dutch DRV for males and females [24,25,26].

#### 2.2.2. Overall Diet Quality

For each patient, two DQIs were calculated from the dietary data, i.e., the Probability of Adequate Nutrient intake (PANDiet) [27] and the Dutch Healthy Diet Index (DHD-index) [28]. Since the PANDiet score is based on nutritional recommendations (i.e., the intake of macro- and micronutrients) and the DHD-index on food based dietary guidelines (i.e., intakes of fruit, vegetables, fish, etc.), the scores were selected to complement each other.

#### 2.2.3. PANDiet

The PANDiet is a nutrient-based DQI which measures the adequacy of intake of 25 macro- and micronutrients in comparison to nutritional recommendations: protein, total fat, saturated fatty acids (SFA), linoleic acid (LA), α-linolenic acid (ALA), the sum of eicosapentaenoic acid (EPA) and docosahexaenoic acid (DHA), total carbohydrate, dietary fiber, vitamins A, B1, B2, B6, B12, C, D, E, calcium, copper, iodine, iron, magnesium, potassium, selenium, sodium, and zinc [29]. Further details of the PANDiet and its validation in the general Dutch population can be found in Appendix A.

#### 2.2.4. DHD-Index

The DHD-index is a DQI that ranks participants according to their adherence to food based dietary guidelines, i.e., in the Netherlands, the Dutch Guidelines for a Healthy Diet of 2006 [30]. These guidelines consist of ten components (i.e., physical activity and the intake of vegetables, fruit (juices), fiber, fish, saturated fatty acids (SFA), trans fatty acids, acidic drinks and foods, sodium and alcohol) which are divided into adequacy components and moderation components. The DHD-index was previously (2012) validated in the general Dutch population [28]. Further details of the DHD-index and its application to the present study can be found in Appendix A.

### 2.3. Data from the General Dutch Population

The 3-day average habitual intake of foods and nutrients in EoE patients, stratified by age group and gender, was compared to reference intakes of males 18 to 69 years (*n* = 1114) and females 18 to 50 years (*n* = 763, according to the classification of the Dutch National Food Consumption Survey (DNFCS, 2007–2010)) [23]. The average dietary assessment in the DNFCS was based on two non-consecutive 24-hour dietary recalls per subject, using the Dutch NEVO-online Food Consumption Database [23]. Nutrient intakes from supplements were not assessed.

### 2.4. Statistical Analyses

Descriptive statistics were used to summarize findings. All data were checked for normality using Shapiro-Wilk tests and histograms. Normally distributed data were presented as means and standard deviations (SD). Skewed data were presented as medians with interquartile ranges (IQR). Categorical data were reported as *n* with percentages of total. To compare dietary intake with the DRV for adults, according to age and sex, skewed data were log-transformed prior to analyses. One-tailed Student’s *t*-tests were performed to test whether dietary intake of the different nutrients were significantly lower (at *p* = 0.05) or significantly higher (at *p* = 0.05) than the reference value (given by the DRV), based on the hypothesis that the intake of the tested nutrient would be unhealthier than recommended. For example: we tested if the fat intake was significantly higher in EoE patients than the maximum recommended intake; however, for most nutrients we tested whether this intake was significantly lower than recommended (some examples are calcium, and vitamins A, C and D). The results were presented as the mean and the upper bound or lower bound of the 95% confidence interval (CI) (if it was tested if the intake of a nutrient was significantly lower and higher than recommended, respectively) on the original (back-transformed) scale.

The DQI scores of EoE patients were compared to the scores of the general Dutch population using an independent two-tailed samples *t*-test. Patient characteristics were compared between the EoE patients and the general Dutch population using independent samples *t*-tests or Mann-Whitney U tests. Food and nutrient intakes between the EoE patients and the general Dutch population were compared using Mann-Whitney U tests. These comparisons were made after the intakes of nutrients were corrected for energy intakes (consumption per 1000 kcal/day), because energy intake in men of 31–50 years differed significantly between the two groups. Statistical analyses were performed using SAS Enterprise Guide v.6.1 and IBM SPSS Statistics v.20.0. *p*-Values of <0.05 were considered statistically significant.

## 3. Results

### 3.1. Patient Characteristics

Baseline characteristics of the 34 EoE patients were presented previously (Table 1, adapted from [15]). In short, the majority of EoE patients (79%) reported having one or more concomitant atopic diseases. Sixty-two percent of the patients reported to have one or more food allergies, including pollen-food syndrome (previously known as Oral Allergy Syndrome). Nineteen patients (56%) avoided specific foods: 12 (35%) because of pollen-food syndrome, 7 (21%) because of food allergy other than pollen-food syndrome and 12 (35%) due to dysphagia, food impaction or dyspepsia. The most frequently avoided food groups were fruit, *n* = 13 (38%), nuts/peanut/seeds, *n* = 13 (38%) and vegetables, *n* = 4 (12%) [15]. Twenty-nine percent of the patients (all male) took dietary supplements, however patients changed brands and types frequently. Age and BMI of EoE patients and the general Dutch population were comparable [23]. However, among EoE patients, a higher percentage was male (76.5% versus 59%, *p* = 0.041).

### 3.2. Comparison to Dietary Reference Values

Table 2 shows that the average percentage of energy (en%) from protein, carbohydrates and total fat in EoE patients was in line with dietary guidelines, although protein and total fat intakes were relatively high. Intake of saturated fat was significantly higher than the DRV (below 10 en%) in males (13.2 en%; *p* < 0.001), whereas dietary fiber intake was significantly lower than the DRV (30 g/d); both in males (19.6 g/d; *p* < 0.001) and females (18.3 g/d; *p* = 0.014). In males, the majority of micronutrient intake levels were in line with the recommended daily amounts with the exception of significantly lower intakes of potassium (3020 vs. 3500 mg/d; *p* = 0.005), magnesium (311 vs. 350 mg/d; *p* = 0.026), selenium (44.5 vs. 60 µg/d; *p* < 0.001), vitamin A (414 vs. 900 µg RE/d; *p* < 0.001) and vitamin D (2.1 vs. 10 µg/d; *p* < 0.001). Females had significantly lower intakes than the DRV of a larger number of nutrients (i.e., iron (10.1 vs. 16 mg/d; *p* = 0.024), sodium (1788 vs. 2400 mg/d; *p* = 0.027), potassium (2794 vs. 3500 mg/d; *p* = 0.020), selenium (34.5 vs. 50 µg/d; *p* < 0.001), vitamins A (405 vs. 680 µg RE/d; *p* = 0.007), B2 (1.3 vs. 1.6 mg/d; *p* = 0.004), C (40.8 vs. 75 mg/d; *p* = 0.009) and D (2.4 vs. 10 µg/d; *p* = 0.002).

### 3.3. Diet Quality Indices

Table 3 shows that the PANDiet and DHD-index of the EoE patients and the general Dutch population were statistically significantly different, with the EoE patients having lower DQI scores compared to the general Dutch population.

### 3.4. Comparison of Intake by EoE Patients with Intake by the General Dutch Population

Male EoE patients had lower energy intakes than the general Dutch population: EoE males (*n* = 26), median 2228 kcal/day, IQR 1709–2556, vs. the general population (*n* = 1114), median 2582 kcal/day, IQR 2131–3088, *p* = 0.003. However, for the different age groups these differences were not statistically significant, except for males aged 18–30 years. For EoE women, there were no statistically significant differences in energy intake.

After correction for energy intake, male EoE patients had a statistically significantly higher energy percentage from fat and had lower intakes of omega-3 fatty acids (EPA and DHA), vitamin D and alcohol per 1000 kcal, while female EoE patients had lower intakes of omega-3 fatty acids, iodine and vitamin C per 1000 kcal than the general Dutch population (Table 4) [23].

In Table 5 it is shown that after correction for energy intake, male EoE patients consumed significantly more vegetables, fruit/nuts/olives, egg/egg products and miscellaneous food groups and, in contrast, less alcoholic beverages and added fat than the general Dutch population. Female EoE patients consumed more potatoes/other tubers, vegetables and fewer non-alcoholic beverages.

## 4. Discussion

EoE is an emerging chronic disease, affecting individuals at any age with a predominance for Caucasian males under the age of 50 [31]. To our knowledge, this is the first study in which the nutritional adequacy and overall diet quality of the habitual diets of adult patients with EoE, prior to any elimination diets other than self-imposed dietary measures, have been assessed by diet scores. In addition, this study is the first in which intake data of adult patients with EoE were compared to those of the general population. In a previous study, we were able to find a relationship between nutrition and the degree of inflammation and mucosal integrity in EoE patients, pointing towards a possible protective effect of a healthy diet consisting of more dietary fiber, fermented dairy and plant-based foods and less fat, animal foods and omega-6-rich oil [15].

In the current study, we showed that for several nutrients, the intake of adult EoE patients did not meet the DRV. Among the nutrients that were statistically significantly different from the DRV, there were several nutrients known for their beneficial or adverse effect on the microbiome, immune system, inflammation or mucosal integrity. For example, the high intake of (saturated) fat in EoE patients is likely to induce a shift in microbiome composition associated with inflammatory processes [32,33]. In addition, intakes of total fat and protein in EoE patients, although within the DRV range, were relatively high, which has been shown to negatively impact the microbiome and inflammatory processes as well [32,34]. These findings point towards a high intake of animal foods, typical for a Western diet. In addition, the low intake of dietary fiber in EoE patients is unfavorable for a healthy gut microbiota [35]. The colonic fermentation of dietary fiber results in the production of short chain fatty acids, which have anti-inflammatory and immune-regulatory benefits [36,37], and are important for the maintenance of the epithelial integrity [38]. Moreover, the vitamin A metabolite, retinoic acid, and the vitamin D metabolite, 1,25-dihydroxyvitamin D3, have direct effects on immune cells, i.e., by enhancing the induction of regulatory T cells and by controlling Th1 and Th17 differentiation [39,40]. Both vitamins A and D, as well as vitamin C, selenium and iron are recognized by the European Food Safety Authority (EFSA) for their local and systemic immunomodulatory properties [41]. Intakes of these nutrients were low in male and/or female EoE patients and may induce a disbalance in their immune system.

Nutritional deficiencies might gain relevance in EoE. There are a few studies on nutritional deficiencies in EoE patients, predominantly in children [42]. In this systematic review, it was found that vitamin D levels of children with EoE, both pre- and post-intervention, were low. One study in adults [43] found that positive skin prick test reaction to peanut was more common in patients who had vitamin D insufficiency (adjusted odds ratio 7.57; *p* = 0.009). However, higher vitamin D levels correlated with higher histologic eosinophil counts (*R* = 0.61; *p* = 0.03).

When comparing the diet composition of our study population to intake levels of the general Dutch population, male EoE patients had significantly lower energy intakes. After correction for energy intake, we found that intakes of several of the nutrients were different between EoE patients and the general Dutch population. EoE patients (males and/or females) had higher total fat intakes and lower intakes of vitamins C and D (all *p* < 0.05). Moreover, EoE patients had lower intake of omega-3 fatty acids. These differences all point towards a diet which has higher pro-inflammatory, lower anti-inflammatory and unfavorable immunomodulatory properties, and overall seems less healthy. Although a significant percentage of EoE patients avoided fruit/vegetables due to pollen-food syndrome, food allergy or EoE symptoms, EoE patients still had a higher relative intake of well-tolerated fruits and vegetables (per 1000 kcal) compared to the general Dutch population. However, fruit and vegetable consumption in both groups was far below (up to 65%) the recommended daily amounts. Remarkably, male EoE patients used less alcohol than male subjects from the general Dutch population. This may be due to the pain and discomfort caused by alcohol in EoE.

Both DQI scores revealed that EoE patients consumed a less healthy diet compared to the general Dutch population. For the PANDiet score, the lower overall diet quality can be explained by differences in nutrient intakes between the two populations (i.e., in total fat, omega-3 fatty acids, iodine and vitamins C and D). The difference in DHD-index score might be explained by differences in the intake of fatty acids. In EoE patients, the intake of total fat was higher, and the intake of omega-3 fatty acids was lower than in the general Dutch population, which led to a lower DHD-index in EoE patients. In contrast to fruits and vegetables that were avoided by half of the EoE patients due to pollen-food syndrome, food allergy or EoE, only few patients (9%) avoided the intake of fish [15], thereby only limitedly contributing to the explanation for the low intake of this food group in the EoE patients.

The differences in DQI scores between the total EoE patient population and the general Dutch population might also partly be explained by a difference in energy intake. Overall, the subjects in the general Dutch population had a higher mean energy intake than the EoE patients, which may have contributed to a better diet quality in the general population as individuals with a higher total energy consumption will meet the requirements for specific nutrients or food groups more easily [17].

The strengths of this study include the use of a detailed, standardized way of recording diet history and the fact that, in addition to the assessment of individual nutrient and food intakes, we used DQI scores to assess the overall diet quality of EoE patients. DQI scores examine the effects of the overall dietary pattern and represent a broader picture of food and nutrient intakes as the combination of foods and nutrients in complex eating patterns and their potential synergistic effects are taken into account. The use of DQI scores may therefore be more predictive of disease risk/severity than individual foods or nutrients. The findings of this study are in line with our previous findings on the relationship between habitual diet and severity of disease in the same study population [15].

Possible limitations of the present study are the small sample size of the EoE population (especially low numbers of females), the fact that we did not take into account the potential influence of self-imposed dietary measures due to food allergy or EoE symptoms on food and nutrient intakes, the differences in timeframes between the dietary assessment of the general Dutch population (between 2007–2010) and the EoE patients (between 2013–2015) and the cross-sectional design of the study. The latter hampers the ability to draw conclusions about the causal relationship between diet quality and disease risk and severity. Large prospective cohort studies or intervention studies are needed to assess if there is a causal relationship between diet quality and EoE. Moreover, the dietary intake of the EoE patients was assessed by a 3-day food record. It is known that the dietary intake of individuals may vary from day to day, and hence the intake of infrequently consumed foods such as fish, may be underestimated.

We did not correct for multiple comparisons in our analyses, due to the explorative character of this study. However, although after correction several significant comparisons will lose significance, the main conclusion of our findings remains intact, namely that the diet of EoE patients has several pro-inflammatory properties, just as the diet of the general Dutch population.

Lastly, as in the DNFCS, we did not calculate the intake of supplements, because of the inconsistent use of amounts and types of supplements by patients, which may have influenced disease outcomes.

In conclusion, intakes of dietary fiber and several micronutrients were below the DRV, while intakes of saturated fat were higher than the DRV in adult patients suffering from EoE. Total protein and total fat intakes were relatively high, yet within the range of the DRV, pointing towards a high intake of animal-based foods. Compared to the general Dutch population, the overall diet of EoE patients, as assessed by two independent diet quality scores, was generally less healthy than the diet of the Dutch population, with the exception of the relative intakes (per 1000 kcal) of vegetables/fruits/olives which were significantly higher (yet still far below (up to 65%) the recommended daily amounts) and the relative intake of alcohol which was significantly lower. Thus, the habitual dietary intake of Dutch adult EoE patients has several pro-inflammatory and unfavorable immunomodulatory properties. These results support the hypothesis that an unhealthy diet is associated with development and progression of EoE. The results of this study are complementary to the results of our recently published cross-sectional study in this population [15].

We are unable to determine whether the diet is changed because of the disease, or that an unhealthy diet precedes the development of EoE. Further prospective and interventional studies are needed to demonstrate causal relationships and effects of diet on the development of EoE. Once the relationship between nutrition and EoE has been established, anti-inflammatory nutrition advice about which foods and nutrients to include and which to avoid could be provided in order to prevent and maintain remission in EoE.

## Figures and Tables

**Table 1 nutrients-13-00214-t001:** Patient characteristics of Eosinophilic esophagitis (EoE) patients (*n* = 34) at baseline.

*n* (%) or Median; IQR	
Male gender in % (*n* = 34)	26/34 (77)
Age in years: median; IQR (*n* = 34)	45.2; 29.0–49.3
BMI (kg/m2): median; IQR (*n* = 34)	24.7; 21.0–26.6
Atopic disease in % (*n* = 34) • Atopic dermatitis • Asthma • Allergic rhinitis • Food allergy ○ of which oral allergy symptoms	27/34 (79) 7/34 (21) 14/34 (41) 17/34 (50) 21/34 (62) 17/34 (50)
Food avoidance in % (*n* = 34) Food groups avoided because of pollen-food syndrome, food allergy or EoEFood groups avoided because of pollen-food syndrome: • one or more types of fruit • one or more types of nuts/peanut/seeds/legumes • one or more types of fruits/vegetables and nuts/peanut/seeds/legumes Food groups avoided because of other food allergy: • one or more types of fruits/vegetables • one or more types of nuts/peanut/seeds • fish • buckwheat • cow’s milk Food groups avoided because of EoE (dysphagia, impaction or dyspepsia): • one or more types of nuts/peanut/seeds • one or more types of fruits/vegetables • alcoholic beverages • dairy • meat • fish • egg • gluten • bread	19/34 (56) 12/34 (35) 2/34 (6) 2/34 (6) 8/34 (24) 7/34 (21) 3/34 (9) 3/34 (9) 1/34 (3) 1/34 (3) 1/34 (3) 12/34 (35) 7/34 (21) 7/34 (21) 5/34 (15) 3/34 (9) 3/34 (9) 2/34 (6) 2/34 (6) 2/34 (6) 1/34 (3)
Use of supplements in % (*n* = 34) • taken on a regular basis • taken on an irregular basis • no supplements • unknown or unclear	8/34 (23) 2/34 (6) 20/34 (59) 4/34 (12)
Baseline measures of eosophageal inflammation (Peak eosinophil count/HPF): median; IQR (*n* = 34)	40; 29–80

IQR—interquartile range.

**Table 2 nutrients-13-00214-t002:** Comparison between habitual diet at baseline and Dutch Dietary Reference Values (DRV) for males 18–69 years and females 18–50 years [24,25,26], except for protein and vitamin B6 where different DRVs apply for 18–50-year-old and >50-year-old males and comparisons are only presented for males and females 18–50 years.

	Males (18–69 Years Old)					Females (18–50 Years Old)				
	*n*	Mean	Upper Bound of 95%CI ^1^	DRV ^2^	*p*-Value ^3^	*n*	Mean	Upper Bound of 95%CI ^1^	DRV ^2^	*p*-Value ^3^
Protein (en%)	21	15.5	16.6	≥8	ns	8	15.5	17.6	≥9	ns
Carbohydrates (en%)	26	41.8	44.8	≥40	ns	8	44.7	52.0	≥40	ns
Fat (en%)	26	37.0	39.6	≥20	ns	8	33.7	38.5	≥20	ns
Saturated fat (en%)	26	13.2	12.3	<10	<0.001	8	12.3	10.0	<10	ns
Dietary fiber (g/day)	26	19.6	21.9	≥30	<0.001	8	18.3	25.7	≥30	0.014
Calcium (mg/day) ^4^	26	874	1063	≥950	ns	8	803	954	≥950	ns
Iron (mg/day)	26	10.3	11.4	≥11	ns	8	10.1	14.5	≥16	0.024
Sodium (mg/day)	26	2628	2941	≥2400	ns	8	1788	2276	≥2400	0.027
Potassium (mg/day)	26	3020	3302	≥3500	0.005	8	2794	3310	≥3500	0.020
Magnesium (mg/day)	26	311	343	≥350	0.026	8	296	402	≥300	ns
Zinc (mg/day)	26	10.2	11.3	≥9	ns	8	9.1	11.1	≥7	ns
Selenium (mcg/day)	26	44.5	48.8	≥70	<0.001	8	34.5	44.6	≥70	<0.001
Iodine (mcg/day)	26	145	167	≥150	ns	8	109	150	≥150	ns
Copper (mg/day)	26	1.1	1.2	≥0.9	ns	8	1.0	1.4	≥0.9	ns
Vitamin A (mcg/day)	26	414	537	≥800	<0.001	8	405	545	≥680	0.007
Vitamin B1 (mg/MJ)	26	0.12	0.13	≥0.1	ns	8	0.10	0.14	≥0.1	ns
Vitamin B2 (mg/day)	26	1.4	1.6	≥1.6	ns	8	1.3	1.4	≥1.6	0.004
Vitamin B3 (mg/MJ)	26	1.9	2.2	≥1.6	ns	8	1.9	2.3	≥1.6	ns
Vitamin B6 (mg/day)	21	1.7	1.9	≥1.5	ns	8	1.4	1.6	≥1.5	ns
Vitamin B12 (mcg/day)	26	3.8	4.4	≥2.8	ns	8	3.5	4.9	≥2.8	ns
Vitamin C (mg/day)	26	74.1	96.2	≥75	ns	8	40.8	59.0	≥75	0.009
Vitamin D (mcg/day)	26	2.1	2.6	≥10	<0.001	8	2.4	4.5	≥10	0.002
Vitamin E (mg/day)	26	11.7	13.4	≥13	ns	8	10.7	14.8	≥11	ns

^1^ One-tailed Student’s t-tests was used to test whether the upper bound of the 95% confidence interval (CI) for daily nutrient intake was significantly lower than the recommended daily amount, except for saturated fat (en%) where it was tested whether the lower bound of the 95% CI for daily intake was significantly higher than the upper level of safe intake; ^2^ For macronutrients, the DRV refers to the reference intake range; for micronutrients, the DRV refers to the recommended daily amount; ^3^
*p*-values < 0.05 were considered statistically significant and are presented in this table, ns—not significant; ^4^ Different DRV of calcium apply for 18–24 years vs. ≥25 years. Due to the low number of patients <25 years (*n* = 1 for males; *n* = 3 for females), DRV for ≥25 years was taken for the whole population.

**Table 3 nutrients-13-00214-t003:** Comparison of Diet Quality Index scores between EoE patients and the general Dutch population.

	EoE			DNFCS 2007–2010			
	*n*	Mean	SD	*n*	Mean	SD	*p*-Value
PANDiet	34	57.0	8.7	1877	61.1	7.3	0.001 *
DHD-index	34	47.1	12.1	1877	50.9	10.4	0.032 *

EoE—eosinophilic esophagitis, DNFCS—Dutch National Food Consumption Survey, SD—standard deviation, PANDiet—Probability of Adequate Nutrient intake, DHD-index—Dutch Healthy Diet Index; *p*-values were obtained from independent two-tailed samples *t*-test. * *p*-values < 0.05 are given in bold.

**Table 4 nutrients-13-00214-t004:** Comparison of nutrient intakes between EoE patients and the general Dutch population for males 19–69 years and females 19–50 years, corrected for energy intake. For protein, calcium and vitamin B6 comparisons are only presented for males and females 19–50 years old. Nutrient intakes are shown per 1000 kcal.

	Males (19–69 Years Old)							Females (19–50 Years Old)						
	EoE			DNFCS 2007–2010				EoE			DNFCS 2007–2010			
	*n*	Median	IQR	*n*	Median	IQR	*p*-Value	*n*	Median	IQR	*n*	Median	IQR	*p*-Value
Protein (en%)	21	16.0	13.0–18.0	763	14.4	12.5–16.6	0.087	8	15.5	15.0–16.0	763	14.7	12.7–16.9	0.431
Carbohydrates (en%)	26	44.0	37.0–49.0	1114	43.9	38.5–49.1	0.622	8	47.5	41.0–49.5	763	47.2	42.1–51.9	0.693
Fat (en%) *	26	38.0	34.0–43.0	1114	34.3	30.5–38.6	0.015*	8	34.0	31.0–38.0	763	33.8	29.1–38.2	0.831
Saturated fat (en%)	26	13.3	11.0–15.0	1114	12.6	10.9–14.6	0.167	8	12.00	10.7–16.0	763	12.6	10.6–14.8	0.978
LA (g/day)	26	6.7	4.5–8.7	1114	6.2	4.7–7.7	0.534	8	7.3	6.2–8.3	763	5.7	4.3–7.4	0.083
ALA (mg/day)	26	747	631–1063	1114	750	583–957	0.542	8	789	683–981	763	682	528–885	0.177
EPA + DHA (mg/1000 kcal)	26	0.0	0.0–0.0	1114	11.8	5.2–43.2	<0.001 *	8	0.0	0.0–20.3	763	13.1	5.5–41.7	0.006*
Dietary fiber (g/1000 kcal)	26	9.0	7.6–10.5	1114	8.6	7.0–10.2	0.174	8	11.3	7.5–12.6	763	9.0	7.3–11.2	0.373
Alcohol (g/1000 kcal)	26	0.0	0.0–2.1	1114	3.0	0.0–10.9	0.002*	8	2.8	0.0–3.3	763	0.0	0.0–3.1	0.244
Calcium (mg/1000 kcal)	22	398	316–479	763	410	298–540	0.998	8	439	391–489	763	468	357–596	0.554
Phosphorus (mg/1000 kcal)	26	640	557–760	1114	676	586–777	0.669	8	621	552–661	763	676	591–768	0.133
Iron (mg/1000 kcal)	26	4.8	4.0–5.9	1114	4.4	3.7–5.3	0.194	8	5.7	4.7–6.8	763	4.7	4.0–5.7	0.126
Sodium (mg/1000 kcal)	26	1246	1013–1384	1114	1186	993–1374	0.359	8	927	721–1332	763	1212	1009–1420	0.077
Potassium (mg/1000 kcal)	26	1436	1202–1610	1114	1467	1238–1691	0.411	8	1509	1441–1617	763	1478	1249–1740	0.638
Magnesium (mg/1000 kcal)	26	145	126–185	1114	148	127–171	0.791	8	171	129–188	763	149	127–178	0.412
Zinc (mg/1000 kcal)	26	4.8	3.9–5.8	1114	4.6	3.8–5.5	0.318	8	4.9	3.9–5.5	763	4.6	3.8–5.5	0.610
Selenium (µg/1000 kcal)	26	19.9	15.7–28.0	1114	19.4	15.8–24.1	0.480	8	18.2	16.3–22.6	763	19.4	16.0–24.2	0.670
Iodine (µg/1000 kcal)	26	68.4	57.1–76.0	1114	76.7	61.1–92.6	0.069	8	64.5	47.5–74.5	763	78.1	63.4–94.8	0.032*
Copper (mg/1000 kcal)	26	0.52	0.44–0.61	1114	0.49	0.42–0.58	0.414	8	0.59	0.49–0.68	763	0.54	0.45–0.63	0.458
Vitamin A (µg/1000 kcal)	26	214	160–280	1114	207	148–290	0.987	8	204	167–261	763	190	134–253	0.480
Retinol Act. Eq. (µg/100 kcal)	26	293	192–387	1114	273	193–400	0.767	8	308	229–455	763	260	192–393	0.401
Vitamin B1 (mg/1000 kcal)	26	0.48	0.37–0.65	1114	0.46	0.35–0.63	0.772	8	0.37	0.33–0.45	763	0.47	0.37–0.64	0.119
Vitamin B2 (mg/1000 kcal)	26	0.69	0.46–0.85	1114	0.66	0.51–0.84	0.868	8	0.67	0.59–0.79	763	0.68	0.53–0.87	0.963
Vitamin B6 (mg/1000 kcal)	22	0.71	0.63–0.89	763	0.79	0.62–1.0	0.422	8	0.79	0.67–0.83	763	0.85	0.64–1.1	0.490
Vitamin B11 (µg/1000 kcal)	26	119	102–170	1114	108	83.9–141	0.076	8	108	106–126	763	115	87.7–145	0.977
Vitamin B12 (µg/1000 kcal)	26	1.6	1.3–2.5	1114	1.8	1.3–2.5	0.497	8	2.0	1.3–3.3	763	1.8	1.3–2.4	0.559
Vitamin C (mg/1000 kcal)	26	35.0	20.5–58.5	1114	34	21–52	0.849	8	22.7	13.4–29.7	763	42.0	27.6–64.7	0.008 *
Vitamin D (µg/1000 kcal)	26	1.1	0.64–1.4	1114	1.4	0.98–1.9	0.011*	8	1.3	0.72–2.8	763	1.3	0.84–1.9	0.859
Vitamin E (mg/1000 kcal)	26	6.0	4.5–6.6	1114	5.4	4.3–6.9	0.640	8	6.3	4.0–7.6	763	5.6	4.5–7.0	0.669

EoE—eosinophilic esophagitis, DNFCS—Dutch National Food Consumption Survey, IQR—Inter Quartile Range, en%—energy percentage, LA—linoleic acid, ALA—α-linolenic acid, EPA—eicosapentaenoic acid, DHA—docosahexaenoic acid; *p*-values were obtained from Mann-Whitney U tests. * *p*-values < 0.05.

**Table 5 nutrients-13-00214-t005:** Comparison of the intake of different food groups (in grams per 1000 kcal) between EoE patients and the general Dutch population.

	Males (19–69 Years Old)							Females (19–50 Years Old)						
	EoE			DNFCS 2007–2010				EoE			DNFCS 2007–2010			
	*n*	Median	IQR	*n*	Median	IQR	*p*-Value	*n*	Median	IQR	*n*	Median	IQR	*p*-Value
Potatoes and tubers	26	29.0	0.0–50.9	1114	39.3	15.3–67.4	0.153	8	55.8	48.6–63.2	763	32.6	0.0–57.0	0.025 *
Vegetables	26	59.3	43.3–83.1	1114	45.5	25.4–71.5	0.011*	8	98.1	73.0–116.7	763	50.6	28.1–82.5	0.007 *
Legumes	26	0.0	0.0–0.0	1114	0.0	0.0–0.0	0.953	8	0.0	0.0–0.0	763	0.0	0.0–0.0	0.480
Fruit, nuts and olives	26	44.1	23.1–88.5	1114	26.6	1.7–62.5	0.031	8	53.3	16.9–71.7	763	38.5	4.6–83.8	0.704
Fruit	26	35.5	9.9–77.4	1114	22.6	0.0–56.1	0.054	8	52.5	16.9–63.2	762	35.7	0.0–80.1	0.634
Dairy products	26	124.1	68.3–191	1114	137.9	70.0–223	0.638	8	126.3	92.0–237.4	763	144.8	82.1–228	0.973
Cereals and cereal products	26	103.7	76.4–122	1114	88.4	63.2–115	0.096	8	76.5	58.8–126	763	93.2	69.3–123	0.493
Meat and meat products	26	50.8	30.2–69.4	1114	46.8	30.3–67.4	0.751	8	35.5	12.1–73.5	763	42.7	23.9–61.6	0.795
Fish and shellfish	26	0.0	0.0–5.2	1114	0.0	0.0–2.9	0.971	8	0.0	0.0–14.9	763	0.0	0.0–1.3	0.860
Eggs and egg products	26	3.7	0.0–15.4	1114	0.0	0.0–7.7	0.046 *	8	0.0	0.0–6.2	763	0.0	0.0–7.7	0.800
Added fat	26	7.2	3.5–11.7	1114	12.0	7.2–16.7	0.007 *	8	9.2	4.6–12.7	763	9.6	5.6–14.4	0.742
Sugar and confectionery	26	9.8	3.4–26.7	1114	14.6	5.0–26.6	0.416	8	17.4	6.4–19.9	763	15.6	6.2–29.7	0.618
Cakes	26	9.6	0.0–22.2	1114	12.0	0.0–26.2	0.605	8	20.8	3.7–34.7	763	17.8	4.8–34.2	0.945
Non–alcoholic beverages	26	473.8	284–735	1114	584.0	435–831	0.115	8	555.6	516–673	763	857.7	651–1208	0.004 *
Alcoholic beverages	26	0.0	0.0–30.8	1114	42.5	0.0–166	0.003 *	8	28.4	0.0–42.1	763	0.0	0.0–28.1	0.171
Condiments and sauces	26	10.2	4.9–16.0	1114	11.1	4.9–20.5	0.478	8	8.6	5.5–12.3	763	10.7	4.5–20.1	0.454
Soups, bouillon	26	0.0	0.0–39.4	1114	0.0	0.0–44.4	0.599	8	0.0	0.0–0.0	763	0.0	0.0–48.5	0.310
Miscellaneous	26	10.9	0.0–26.0	1114	0.0	0.0–8.5	<0.001 *	8	0.0	0.0–7.1	763	0.0	0.0–10.6	0.821

EoE—eosinophilic esophagitis, DNFCS—Dutch National Food Consumption Survey, IQR—Inter Quartile Range; *p*-values were obtained from Mann-Whitney U tests. * *p*-values **<** 0.05.

## Data Availability

The data presented in this study are available on request from the corresponding author. A data sharing agreement will be requested.

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
