# Peer review of "The Habitual Diet of Dutch Adult Patients with Eosinophilic Esophagitis Has Pro-Inflammatory Properties and Low Diet Quality Scores"

_nutrients, 2021, doi:10.3390/nu13010214_

Round 1

Reviewer 1 Report

General Comments

This is a very interesting paper, specifically evaluating the diet of patients with EoE and comparing the results with both those obtained from the general population and recommended reference values for the Dutch population. It was very well written and had a very sound methodological approach.

Introduction

Most of the introduction discussed what the authors have previously shown or described the methods. The introduction should have focussed more on the condition itself, dietary approaches to EoE and also the fact that many patients with EoE do also have pollen-related food allergy, which is relevant for this paper. There could have also been more discussion about how EoE patients often modify their diet by avoiding certain foods and eating more slowly.

Line 53 on p 2 – there is a spelling mistake – the word should be threefold not tree-fold.

Materials and Methods

This was overly long and very detailed, especially the section on the PANdiet and the DHD index. For example, the paragraph 2.2.4 could have been one line added to section 2.2.3.  Much of the information could be in supplementary files. The statistical analysis section was also overlong.

Results

General point – I prefer the tables to be at the end of the document rather than inserted between each section

Section 3.1 - It would have made the paper more interesting to state in more detail which foods were being avoided - a bar chart giving percentages of foods being avoided would have enhanced this section. Some detail on food patterns would have also been valuable. Although the dietary supplements changed frequently, they should have been reported as part food the 3 day food diary.

Section 3.2 – this was very difficult to read and also repeated entirely what was on Table 1. There is no need to repeat all of the upper value 95%CI detail in this paragraph. This should have been a much more descriptive narrative, focussing on the main differences.

Table 1a and 1b – Table 1 b seemed to be entirely a repeat of data in Table 1a.

Discussion

The discussion was good but could have again included more about the fact that EoE often affects more males than females and also younger age groups. It should also have focussed on the fact that often due to dysphagia, some foods are avoided altogether, and more discussion could have been focussed on the avoidance of fruits and nuts due to pollen-related food allergy. However a lot of assumptions were made regarding food and nutritional intakes when actually for most nutrients, there was no difference between EoE patients and age-matched general population. The supplements taken by some participants were not included in the dietary calculations and this is acknowledged in the limitations but it would have been helpful to know whether it is more males than females who took supplements and whether there are other lifestyle differences that might explain the low levels of some nutrients. For example, fish was less likely to be consumed and this might explain low levels of omega 3 fats but could the avoidance of fish be due to the EoE – this could have been discussed.

There was some repetition with lines 279-280 and 282-283 which could be merged.

Line 268 states that EoE patients had a higher intake of saturated fat whereas there appeared to be no significant difference between EoE patients and the general population according to Table 4 A.

Line 270 – states total fat and protein intakes were high although within the DRV – but again they were no different to those of the general population.

Reviewer 2 Report

A very well written paper on a relevant topic, EoE, with easily applicable intervention into the patients´diet suggesting very favourably therapeutic effect, sparing corticosteroids and other medications that may also cause side effects in long term.

Reviewer 3 Report

de Kroon et al. in their manuscript entitled “The habitual diet of Dutch adult patients with Eosinophilic Esophagitis has low diet quality scores and pro-inflammatory properties” report an interesting study investigating the role of diet in EoE patients. The study provides innovative data about the quality of diet in EoE patients and it is well developed. I have just some minor suggestions, which are as follows:

I understand the study group is limited and the data retrospectively collected in patients that participated in previous trials. However, I wonder whether diet limitation in EoE patients due to allergies or intolerance might influence body composition and body health. In addition, I understand the Authors' should somehow limit their consideration about pro inflammatory state considerations since they did not evaluate this condition in the patients. However, there are evidence that nutritional deficiencies might gain relevance in EoE and a reference to it in the discussion would be much welcomed.

Just for my curiosity, out of all patients, do the authors have some information related body composition? Total fat mass, fat-free mass, trunk fat mass and body fat percent? Could be very interesting to analyze these data to evaluate a relationship between diet and effect in body health. Furthermore, is data about serum levels of cholesterol, glucose, triglyceride, micronutrients ... available to authors to investigate a relationship between the quantity of macro and micronutrients intake and serum values?
